# Multimodal Investigation of Deep Gray Matter Nucleus in Patients with Multiple Sclerosis and Their Clinical Correlations: A Multivariate Pattern Analysis Study

**DOI:** 10.3390/jpm13101488

**Published:** 2023-10-12

**Authors:** Feiyue Yin, Zichun Yan, Yongmei Li, Shuang Ding, Xiaohua Wang, Zhuowei Shi, Jinzhou Feng, Silin Du, Zeyun Tan, Chun Zeng

**Affiliations:** 1Department of Radiology, The First Affiliated Hospital of Chongqing Medical University, Chongqing 400016, China; 2021120293@stu.cqmu.edu.cn (F.Y.); 2022140087@stu.cqmu.edu.cn (Z.Y.); 202071@cqmu.edu.cn (Y.L.); 2022130151@stu.cqmu.edu.cn (X.W.); 2021110183@stu.cqmu.edu.cn (Z.S.); 2021440245@stu.cqmu.edu.cn (S.D.); 2021150058@stu.cqmu.edu.cn (Z.T.); 2Department of Radiology, The Childrens’ Hospital of Chongqing Medical University, Chongqing 400015, China; 2021140110@stu.cqmu.edu.cn; 3Department of Neurology, The First Affiliated Hospital of Chongqing Medical University, Chongqing 400016, China; 203756@cqmu.edu.cn

**Keywords:** magnetic resonance imaging, multiple sclerosis, deep gray matter, diffusion tensor imaging

## Abstract

Deep gray matter (DGM) nucleus are involved in patients with multiple sclerosis (MS) and are strongly associated with clinical symptoms. We used machine learning approach to further explore microstructural alterations in DGM of MS patients. One hundred and fifteen MS patients and seventy-one healthy controls (HC) underwent brain MRI. The fractional anisotropy (FA), mean diffusivity (MD), quantitative susceptibility value (QSV) and volumes of the caudate nucleus (CN), putamen (PT), globus pallidus (GP), and thalamus (TH) were measured. Multivariate pattern analysis, based on a machine-learning algorithm, was applied to investigate the most damaged regions. Partial correlation analysis was used to investigate the correlation between MRI quantitative metrics and clinical neurological scores. The area under the curve of FA-based classification model was 0.83, while they were 0.93 for MD and 0.81 for QSV. The Montreal cognitive assessment scores were correlated with the volume of the DGM and the expanded disability status scale scores were correlated with the MD of the GP and PT. The study results indicated that MS patients had involvement of DGM with the CN being the most affected. The atrophy of DGM in MS patients mainly affected cognitive function and the microstructural damage of DGM was mainly correlated with clinical disability.

## 1. Introduction

In multiple sclerosis (MS) patients, detailed pathological examination has revealed widespread pathology in the deep gray matter (DGM) [1]. Atrophy and microstructural changes in DGM regions, which have been well-documented and clinically significant in MS using MRI [2,3], are not fully understood in terms of the exact underlying mechanisms, potentially indicating complex interactions among various aspects of MS pathology.

Iron deposition in DGM is common in MS patients, which may be related to phagocytosis of dead myelin and oligodendrocyte debris by macrophages and T cells crossing the blood-brain barrier, and release of bound iron from the cells and myelin sheaths, which in turn leads to increased iron in the SN [4]. Quantitative susceptibility mapping (QSM) allows non-invasive quantitative assessment of iron content in brain tissue using quantitative susceptibility value (QSV) [5,6]. Diffusion MRI parameters can estimate microstructural changes associated with diffusion anisotropy, such as fractional anisotropy (FA) and mean diffusivity (MD), which can reflect the degree of microstructural organization and coherence in tissues. However, previous studies have mostly explored diffusion changes in white matter and its clinical relevance in MS patients, but fewer studies have examined that in DGM [7,8]. Early atrophy of DGM in MS patients is closely related to clinical prognosis and previous studies have demonstrated multiple mechanisms, including mitochondrial failure, iron deposition, retrograde degeneration through white matter lesions [9], and network overload and collapse in the cingulate gyrus and precuneus cortex [10], which ultimately leads to a reduction in the number of DGM cells, further leading to atrophy of DGM.

Multivariate pattern analysis (MVPA) is an analytical technique in imaging studies that employs machine learning. It enables the simultaneous analysis of the spatial patterns in voxel signal formations utilizing a larger dataset for enhanced sensitivity and improved classification accuracy in categorizing individuals [11,12]. While MVPA has gained traction in studying neuropsychiatric disorders like epilepsy and Alzheimer’s disease [13,14], the application in MS patients remains uncommon. The specific weights of corresponding features for the classification model can be obtained, which is beneficial for exploring the most damaged regions in DGM of MS patients.

We assume that DGM still exhibit widespread involvement in MS patients without lesions existence within the DGM regions with varying degrees of involvement in each nucleus of DGM, which is correlated with cognitive impairment and disability progression. Therefore, the tasks of the study were to (i) apply DTI and QSM to quantitatively evaluate the degree of microstructural damage and iron accumulation in a cohort of MS patients and HC, (ii) incorporate FA, MD, and QSV into a machine learning model based on support vector machine (SVM) to investigate the most damaged areas in MS patients and (iii) explore relationship between MRI quantitative metrics of DGM in MS patients and clinical neurological scales.

## 2. Materials and Methods

### 2.1. Participants and Study Procedures

#### 2.1.1. Sample Collection

Two hundred MS patients and seventy-one healthy controls (HC) were enrolled from the Department of Radiology, the First Affiliated Hospital of Chongqing Medical University. All the participants were recruited between May 2021 and January 2023. Inclusion criteria for MS patients were as follows: (1) a confirmed diagnosis of MS according to the 2017 revised McDonald’s diagnostic criteria [15], (2) age 18–60 years, and (3) absence of neurological conditions other than MS. The exclusion criteria were as follows: (1) patients with contraindications for MRI scans, (2) history of intravenous corticosteroid treatment within 2 months before the imaging examinations, (3) image artifacts or incomplete clinical information, and (4) 3-dimensional fluid-attenuated inversion recovery (FLAIR) hyperintense lesions existence in DGM regions.

#### 2.1.2. Clinical Neurological Evaluation

The mini-mental state examination (MMSE), the Montreal cognitive assessment (MoCA), and the symbol digit modalities test (SDMT) scores were used to assess the cognitive performance of all participants. The expanded disability status scale (EDSS) scores were used to assess the clinical disability of patients.

#### 2.1.3. Research Design

Some patients were excluded due to the lack of DTI or QSM sequences or clinical neurological scales or the existence of the lesions in DGM regions, and ultimately, one hundred and fifteen MS patients and seventy-one HC were included in study 1. As there were differences in age and sex between the patient group and the HC group in study 1, we performed comparisons after a 1:1 nearest-neighbor propensity score matching, including age, sex and education duration as covariates. As a result, fifty-three MS patients and fifty-three HC were included in study 2. Finally, we conducted MVPA on above two groups based on registered FA, MD, and QSM images using the PRoNTo software (http://www.mlnl.cs.ucl.ac.uk/pronto accessed on 30 September 2023). Due to the suboptimal registration performance of QSM, we excluded patients with failed registered QSM and corresponding PSM matches. As a result, only thirty MS patients and thirty HC were included in study 3 (Figure 1). All patients included in this study were relapsing-remitting MS patients.

### 2.2. MRI Protocols

All MR scans were performed on a 3-T MR scanner (Magnetom Skyra, Siemens Healthcare GmbH, Erlangen, Germany) using a 32-channel head coil. A standard protocol for MS studies included a sagittal 3-dimensional T1-weighted magnetization prepared rapid gradient echo (3D-T1 MPRAGE) sequence [echo time (TE) = 2.26 ms, repetition time (TR) = 2300 ms, inversion time (TI) = 900 ms, 192 slices, field of view (FOV) = 256 mm, voxel size = 1.0 × 1.0 × 1.0 mm, acquisition time (TA) = 5:21 min] and a sagittal 3D-FLAIR (TE = 388 ms, TR = 5000 ms, TI = 1800 ms, 192 slices, FOV = 256 mm, voxel size = 0.5 × 0.5 × 1.0 mm, TA = 7:07 min). DTI data were acquired using the following parameters: TE = 97 ms, TR = 5000 ms, 25 slices, FOV = 220 mm, voxel size = 1.7 × 1.7 × 2.0 mm, TA = 6:04 min, integrated Parallel Acquisition Techniques (iPAT) acceleration factor = 2 (GRAPPA), Partial-Fourier = 6/8, and three b values (0, 1000, and 2000 s/mm^2^) with diffusion encoding in 30 directions. QSM data were acquired using the following parameters:TE1 = 7.5 ms, ∆TE = 7.5 ms, TE4 = 30.0 ms, FOV = 220 mm × 220 mm, Matrix = 256 × 256, Thickness = 2.0 mm, FA = 20°.

### 2.3. MRI Image Processing

#### 2.3.1. Volumetric Measurements

Lesions were segmented by the lesion growth algorithm (LGA) as implemented in the lesion segmentation toolbox (LST) version 3.0.0 (www.statisticalmodelling.de/lst.html, accessed on 30 September 2023) for Statistical Parametric Mapping (SPM) 12 in MATLAB R2013b (Version 8.2.0.701) based on 3D-T1 MPRAGE. After visual inspection and manual correction, lesions were filled on 3D-T1 images. The volumetric segmentation of DGM structures were performed with the FreeSurfer image analysis suite (http://surfer.nmr.mgh.harvard.edu/, accessed on 30 September 2023) using lesion-filled 3D-T1 images as input. Firstly, the raw data underwent several pre-processing steps to eliminate unwanted noise, artifacts, or other irregularities. Following this, automated segmentation (algorithmsASegStatsLUT) was utilized to the DGM into CN, PT, GP and TH followed by the calculation of their volumes. Meanwhile, the total intracranial volume (TIV) of each subject was also obtained.

#### 2.3.2. QSM Reconstruction

QSM reconstruction was performed using the Sepia platform based on MATLAB R2013b (Version 8.2.0.701) (sepia-documentation.readthedocs.io). The reconstruction steps were followed: (1) Preprocessing, which involved zero-padding the k-space data along the phase encoding direction to achieve isotropic resolution [16]. (2) Phase unwrapping, where the Laplacian method was employed to unwrap the data with phase wrapping [17], effectively preserving the low-frequency components of all brain tissues (e.g., gray matter, white matter, cerebrospinal fluid). (3) Background field removal, accomplished by mitigating the background field using complex harmonic artifact reduction on phase data, as the unwrapped phase map still contained background field that masked the brain tissue phase information. (4) Artifact removal, employing Star-QSM to eliminate streaking artifacts while preserving clear anatomical details. (5) Susceptibility inversion, utilizing a morphology-constrained dipole inversion algorithm to convert the data into a susceptibility map. These steps resulted in the reconstruction of the original k-space data into a QSM image. To minimize additional inter-subject heterogeneity, the average QSV of the brain was used as a reference by considering a larger reference region. All QSM images were individually registered to their T1 images and subsequently registered to the Automated Anatomical Labeling 3 (AAL3) template using FSL to extract QSV for DGM. The registered QSM images were utilized for subsequent MVPA in study 3.

#### 2.3.3. DTI Constructions

The DTI data underwent several pre-processing steps, which included denoising [18], removal of Gibbs ringing artifacts [19] and correction of subject motion [20] using the MRtrix3 (3.0.4) [21] package. Additionally, FMRIB Software Library (FSL) (https://fsl.fmrib.ox.ac.uk/fsl/fslwiki/, accessed on 30 September 2023) was utilized to correct for eddy-currents [22] and susceptibility-induced distortions [23]. The diffusion parametric maps including FA and MD were computed using the diffusion kurtosis estimator software (DKE) (https://www.nitrc.org/projects/dke/, accessed on 30 September 2023), which then registered to AAL3 template. The diffusion tensor parametric maps, specifically FA and MD, were computed using DKE. The DKE employed the constrained linear least-squares quadratic programming (CCLS-QP) algorithm with standard parameters including spatial smoothing and strong median filtering. The specific parameters used with DKE were as follows: a Gaussian kernel with a full width at half maximum (FWHM) of 3.375 mm, constraint on directional kurtosis (Kmin = 0, NKmax = 3, and D > 0), and thresholds on output kurtosis maps (0 < K < 3). Constrained linear weighted fitting was applied for DTI fitting to generate the DTI-based parametric maps. Finally, DGM binarized masks obtained based on AAL3 template were multiplied with the corresponding registered FA and MD maps to extract FA and MD values for DGM. The registered FA and MD images were utilized for subsequent MVPA in study 3.

#### 2.3.4. Multivariate Pattern Analysis

MVPA was performed via the PRoNTo Toolbox (http://www.mlnl.cs.ucl.ac.uk/pronto, accessed on 30 September 2023) to explore the most prominent regions of microstructural damage and iron deposition in MS patients compared with HC based on FA, MD and QSM images. The main steps included (1) feature selection; (2) training the SVM classifier; (3) evaluating the classifier model; (4) calculating the weights of brain regions. An SVM seeks a separating hyperplane that optimizes the distinction between two classes. This is achieved by first mapping the 3D image input data into a high-dimensional feature space. A gaussian kernel SVM was used to construct a nonlinear classifier with selected radiomics features. Finally, the precision, sensitivity and specificity of FA-based models, MD-based models and QSV-based models were obtained. To validate SVM classifiers, 1000 permutation tests were used. Besides, the receiver operating characteristic (ROC) curve and the area under the curve (AUC) were calculated to evaluate the performance of the classifier model. For region of interest extraction and selection, DGM masks obtained based on AAL3 template was utilized to compute weights of each nucleus in this study.

The Gaussian kernel function, also known as the Radial Basis Function (RBF), which is introduced to capture nonlinear relationships between input features by mapping them into a higher-dimensional space using the formula:*K*(*x, y*) *= exp*(−γ ||*x − y*||^2^) 

### 2.4. Statistical Analysis

Statistical tests were conducted using SPSS V.26.0 (IBM Corporation, New York, NY, USA). Kolmogorov–Smirnov test was used to analyze the normality of metric data. Normally distributed metric data was represented as $\bar{x}$ ± s, while non-normally distributed metric data was represented as M (Q1, Q3).

Continuous data were compared between MS patients and HC using independent samples *t* test or Mann-Whitney U test. Categorical outcomes were evaluated using χ2 test. The volumes between the two groups of participants were compared using with age, sex, education duration and TIV as covariates and the FA, MD and QSV of CN, PT, GP and TH were compared with age, sex, education duration and the volumes of corresponding nucleus serving as covariates using multivariate analysis of variance. Evaluation of the classification accuracy of MS and HC using QSV, FA and MD images of the DGM as predictors were based on MVPA—supported SVM classifier.

Partial correlation analysis was used to assess the correlation between the volume, QSV, FA, MD of DGM and clinical neurological scores after adjusting for age, sex and education duration including all the patients in study 1 (one hundred and fifteen MS patients). All clinical neurological scores were transformed into Z-scores for comparison. Due to the exploratory nature of the study, no adjustment for multiple comparisons was made. A *p*-value of <0.05 was considered statistically significant.

## 3. Results

### 3.1. Demographics and Clinical Neurological Scores of MS Patients and HC Groups in Study 1 and Study 2

The demographics and clinical neurological scores of patients with MS and HC in study 1 are summarized in Table 1. The MS group were younger (*p* < 0.001) and had a lower proportion of male participants (*p* < 0.045) compared to HC. In MS patients, both MMSE scores (*p* < 0.001) and MoCA scores (*p* = 0.027) were lower compared to the HC. There were no significant differences in education years (*p* = 0.771) and SDMT scores (*p* = 0.954) between the two groups (Table 1). Among the 115 MS patients, 38 participants used Teriflunomide, 28 used Siponimod, 5 used Fingolimod.

There were no statistically significant differences in age (*p* = 0.054) and sex (*p* = 0.636) between MS patients and HC in study 2 (Table 2).

### 3.2. Comparison of MRI Quantitative Metrics between MS Patients and HC in Study 1 and Study 2

In study 1, the volumes of DGM were significantly lower in MS group compared to HC group (*p* all < 0.001). The QSV of CN, PT, GP in MS group were higher than those in HC group (*p* all < 0.001). However, QSV values of TH in MS patients were lower than in the HC group (*p* < 0.001). The FA of the CN was lower (*p* < 0.001) than that in HC group and the MD of the CN (*p* = 0.007) and PT (*p* = 0.020) were higher than that in HC group (Table 1, Figure 2). In study 2, after adjusting for the effects of age, sex and volume, the results were similar (Table 2).

### 3.3. MVPA Evaluation

Comparison in DGM regions between MS patients and HC group were based on FA, MD and QSV images. The accuracy (permutation *p* = 0.001, 1000 times), sensitivity and specificity of FA-based model were 79.66%, 76.67% and 80.00%, with an AUC of 0.83. The accuracy (permutation *p* = 0.001, 1000 times), sensitivity and specificity of MD-based model were 81.36%, 83.33% and 76.67%, with an AUC of 0.93. The accuracy (permutation *p* = 0.002, 1000 times), sensitivity and specificity of QSM-based models were 69.49%, 73.33% and 63.33%, with an AUC of 0.81. The ROC and prediction plots of above three model are shown in Figure 3 and Figure 4. The most informative regions for discrimination based on FA-based models was CN, which exhibited the highest difference, followed by GP, PT and TH. Similarly, in terms of MD-based models, the CN exhibited the greatest difference, followed by GP, TH and PT. Furthermore, when comparing QSM-based models, the CN displayed the largest difference, followed by PT, GP and TH (Table 3, Figure 5).

### 3.4. Correlations between MRI Quantitative Metrics and Clinical Neurological Scores in MS Patients

The volume of the GP (*r* = 0.289, *p* = 0.049), TH (*r* = 0.365, *p* = 0.012) and the MD of the TH (*r* = −0.419, *p* =0.003) showed a correlation with the SDMT scores. The volume of the PT (*r* = 0.33, *p* = 0.024), GP (*r* = 0.343, *p* = 0.018) and TH (*r* = 0.324, *p* = 0.026) showed a correlation with MOCA scores. The MD values of the PT (*r* = −0.331, *p* = 0.023) and GP (*r* = −0.478, *p* = 0.001) were correlated with the EDSS scores. The volume of GP (*r* = −0.477, *p* = 0.001), TH (*r* = −0.406, *p* = 0.005) and MD of TH (*r* = 0.427, *p* = 0.003) were significantly correlated with disease duration. No correlation was found between the imaging parameters and MMSE scores (Figure 6).

## 4. Discussion

This study employed high-resolution, quantitative and multi-parametric MRI methods to investigate morphometrics, iron concentration variations and diffusion metrics of the DGM in MS patients to investigate the microstructural changes in DGM in the absence of lesions. We also explored the relationship between MRI quantitative parameters showing microstructural damage and clinical neurological scores in order to better understand the pathophysiological mechanisms of MS patients and thus better serve the clinic.

Firstly, we investigated the volume changes of DGM in patients with MS and found that the volumes of CN, PT, GP and TH were significantly reduced, consistent with previous research findings [21,24,25]. Our study results also demonstrated a significant correlation between the atrophy of DGM in MS patients with MoCA and SDMT, which has been supported by previous studies [22,23]. This may be attributed to progressive gray matter atrophy resulting from neuronal loss, axonal damage and synaptic dysfunction over the course of the disease, further contributing to decline in cognitive function in patients.

In the meanwhile, a statistically significant decrease in magnetic susceptibility in the TH of MS patients was also observed. It is worth discussing whether the decrease in thalamic magnetization is a result of thalamic atrophy or a reduction in iron content. We hypothesize that it is related to a decrease in iron content. In this study, despite correcting for thalamic volume, a decreased magnetic susceptibility in the TH of MS patients was still observed, suggesting a link between the decreased thalamic magnetization and reduced iron content. In fact, MS patients often exhibit decreased iron content. Du et al. [26] in a 2-year longitudinal study, found a significant decrease in iron content of substantia nigra. Haider et al. [27], based on histopathological findings, indicated that iron is primarily stored in oligodendrocytes and myelin fibers and released during demyelination. His study also demonstrated that iron deposition contributes to the diffuse neurodegeneration, which may perpetuate a state of chronic inflammation, leading to further oligodendrocyte damage and iron depletion, resulting in the loss of iron-containing cells such as oligodendrocytes and neurons, which may be one of the reasons for the decrease iron content in TH in our research.

For basal ganglia regions, previous studies have shown varying degrees of increased magnetic susceptibility in patients with MS. Al-Radaideh et al. [28] reconstructed iron mapping based on T2*-weighted revealed increased magnetization in the CN and GP. Taege et al. [29] found increased magnetic susceptibility in the TH, CN and GP based on the iron microstructural coefficient. Hagemeier et al. [30] identified increased magnetic susceptibility in the TH and GP using QSM. In summary, most studies indicated an increased magnetic susceptibility in basal ganglia of MS patients, which possibly due to the phagocytosis of demyelinated and apoptotic debris by macrophages and T cells crossing the blood-brain barrier, leading to the release of iron bound within cells and myelin. This, in turn, results in increased iron content in DGM regions [31]. Discrepancies with previous articles may be attributed to sample heterogeneity and variations in iron sensitivity across different acquisition sequences.

Previous studies have primarily focused on microstructural damage in the cortical gray matter and white matter with limited research on DGM. Furthermore, the results were inconsistent. Some studies [32,33] have found no differences in FA and MD of the DGM between MS patients and HC. Solana et al. [34] found increased FA in the PT of MS patients, but decreased MD in the right GP. Ciccarelli et al. [35] found increased FA values in the TH, CN and PT, with decreased MD in the PT. These differences can be attributed to differences in DTI acquisition parameters as well as post-processing, different disease duration of the included patients. Hasan et al. [36] found increased MD in the TH based on diffusion tensor imaging (DTI), which may be a result of excessive dendritic growth and disorganized branching structure in the TH. In this study, there is a trend of increased FA and decreased MD in DGM regions of MS patients. Previous positron emission tomography (PET) studies investigating regional glucose metabolism have reported basal ganglia hypometabolism, indicating remote functional metabolic effects (neurodegeneration) of the fiber connections [37]. This disruption in metabolism can result in constrained diffusion as observed in studies on focal ischemia and status epilepticus [38], which can be influenced by various pathological processes like inflammation-related edema, axonal loss, gliosis [39]. Hence, the overall pattern of decreased FA and increased MD may indicate a combination of axonal loss and gliosis.

The uniqueness of this study lied in the use of machine learning methods to compare the differences in microstructural changes of the DGM between MS patients and HC aiming to identify the most damaged regions. Due to the small sample size, all models underwent 1000 permutation tests to ensure reliability and generalizability. Interestingly, CN had the highest weight in all three models classifying MS patients and HC. Combining the above research findings, we found that compared to HC, MS patients showed the most pronounced microstructural changes in CN. As a part of the DGM, the CN is involved in fine motor and cognitive functions [34], which may be a contributing factor to the decline in clinical cognitive function in some MS patients without lesion formation. More studies are needed in the future to explain and confirm the exact mechanism of the severe microstructure damage of CN in MS patients.

The TH is a crucial hub region of the brain. Previous studies have shown that MS patients have extensive thalamic atrophy, which is strongly associated with clinical cognition [40,41]. However, the TH had the lowest weight in the three models of FA, MD and QSV that compared MS patients with HC in the MVPA analysis, indicating that correlations between TH and clinical symptoms are primarily driven by thalamic volume rather than thalamic microstructural damage. The thalamic atrophy was severe but the thalamic microstructural damage was mild in this study, which may be due to the fact that the brain has a certain degree of plasticity and compensatory capacity and when one region is damaged, other regions may enhance their function to compensate for the loss of function, thus maintaining relatively normal function with less severe microstructural damage. In addition, DMT treatment, as well as neuroprotective mechanisms, may help to counteract the damage caused by inflammation and demyelination.

This study found a negative correlation between MD of the GP and PT and EDSS in MS patients, indicating that microstructural damage in DGM contributes to disability. However, our study did not find any association between EDSS and the volume or iron deposition in DGM as other studies have reported [42,43], which may be attributed to a relatively small sample and short disease duration as well as differences in segmentation methods and voxel sizes resulting in variations in anisotropy.

The main limitation of this study is the relatively small sample size and further research is needed to validate these findings by studying a larger sample of MS patients and including a broader range of MS phenotypes. Furthermore, it is important to note that this study is cross-sectional in nature and relies on data from a single imaging time point, which limits our understanding of how the parameters evolve over time. In addition, lesion burden has a significant impact on MS patients. However, this study did not include an investigation of the correlation between lesion burden and clinical neurological scales.

## 5. Conclusions

In summary, our study confirmed the presence of volume atrophy and microstructural change in DGM without lesions in MS patients. Among them, CN had the most obvious microstructural damage and TH had the least. The atrophy of DGM in MS patients significantly affected cognitive function and the microstructural damage were correlated with clinical disability in MS patients.

## Figures and Tables

**Figure 1 jpm-13-01488-f001:**
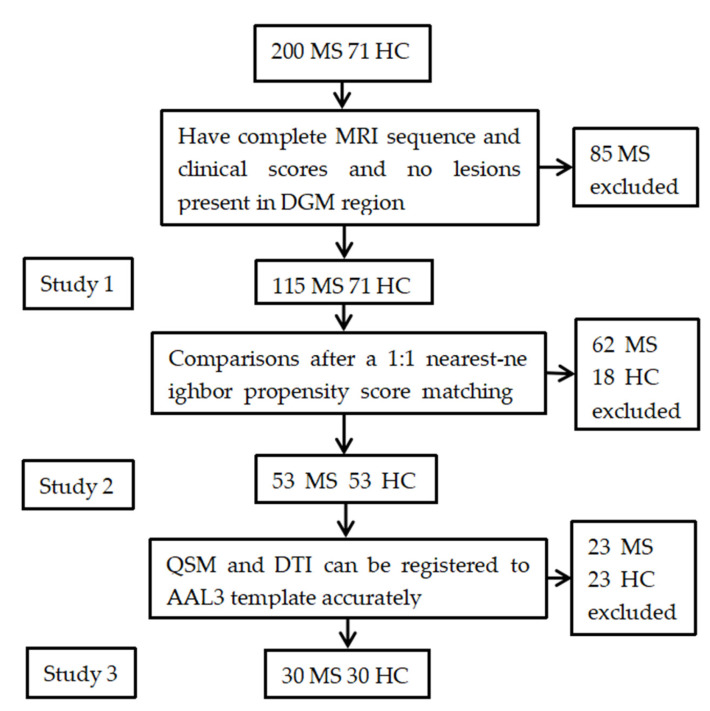
Flow chart and study design shows how we enrolled patients with MS patients and HC. MS, multiple sclerosis; HC, healthy control; DTI, diffusion tensor imaging; QSM, quantitative susceptibility mapping; SVM, support vector machine.

**Figure 2 jpm-13-01488-f002:**
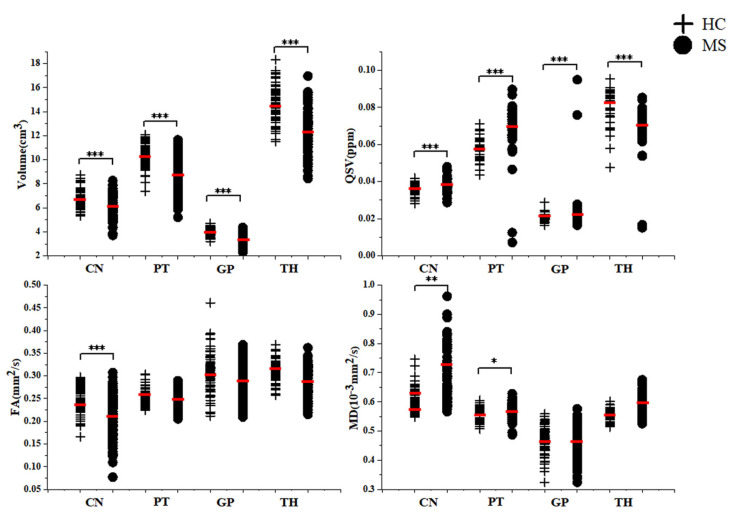
Plots of differences in volume, QSV, FA and MD values in DGM between MS patients and HC group in study 1 corrected for age, sex, education duration and corresponding nucleus volume; QSV, quantitative susceptibility value; FA, Fractional anisotropy; MD, Mean diffusivity; CN, caudate nucleus; PT, putamen; GP, globus pallidus; TH, thalamus; MS, multiple sclerosis; HC, healthy control; * <0.05; ** <0.01; *** <0.001.

**Figure 3 jpm-13-01488-f003:**
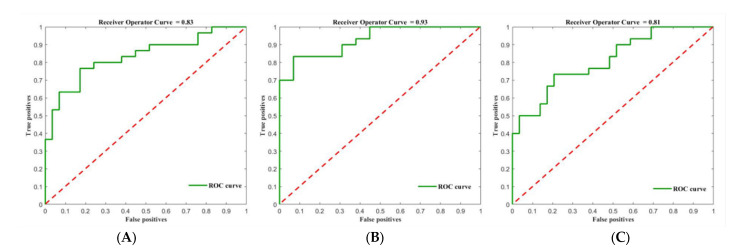
The ROC of classifier of FA-based (**A**), MD-based (**B**) and QSV-based (**C**) models. FA, fractional anisotropy; MD, mean diffusivity; QSV, quantitative susceptibility values.

**Figure 4 jpm-13-01488-f004:**
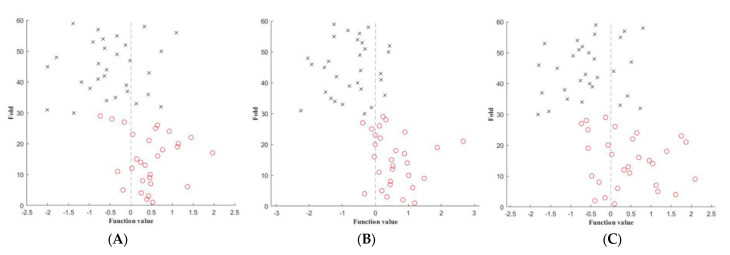
The prediction of MVPA classification between MS patients and HC on FA (**A**), MD (**B**) and QSV (**C**) derived from DTI and QSM. FA, fractional anisotropy; MD, mean diffusivity; QSV, quantitative susceptibility values.

**Figure 5 jpm-13-01488-f005:**
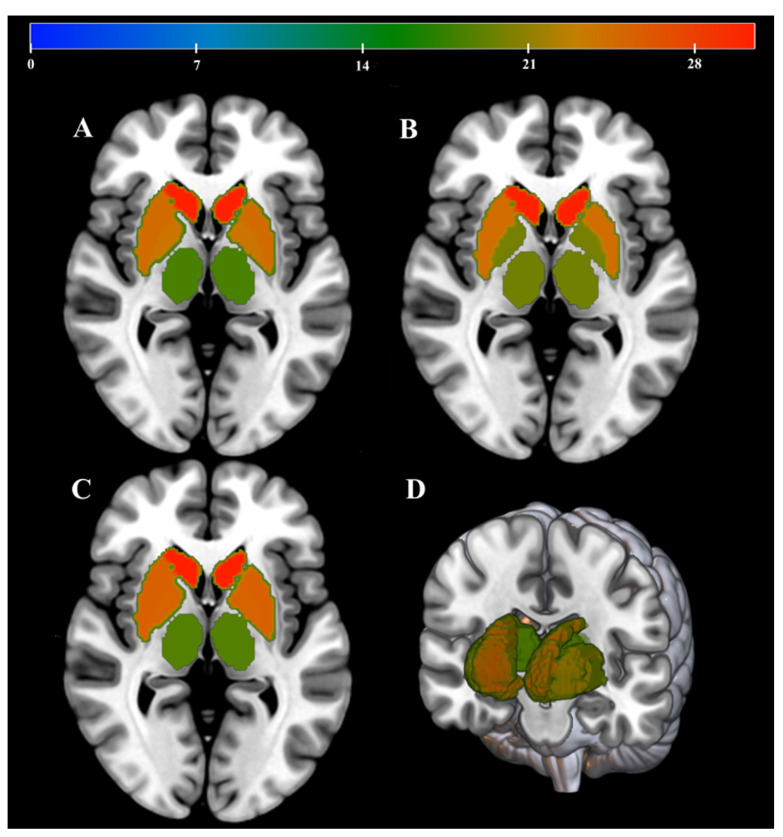
Schematic diagrams of the differences in FA, MD and QSV of the DGM of MS patients versus HC based on MVPA. (**A**–**C**) correspond to the weighting diagrams of the differences in FA, MD and QSV respectively. The redder color represents the larger difference between MS patients and HC in the corresponding region. (**D**) shows a 3D map of DGM.

**Figure 6 jpm-13-01488-f006:**
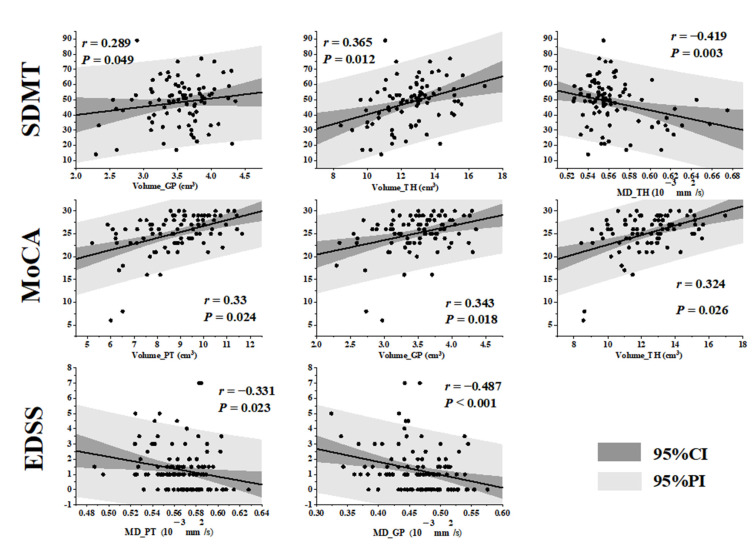
Scatter plot of correlation between MRI quantitative parameters and SDMT, MoCA and EDSS scores in DGM of MS patients.

**Table 1 jpm-13-01488-t001:** Demographic and clinical neurological scores and MRI quantitative metrics comparisons between MS patients and HC in study 1.

		HC (*n* = 71)	MS Patients (*n* = 115)	*p*
**Sex (male/female)**		15/36	12/72	0.045 ^a^
**Age (years)**		39.8 (30,49)	32.87 (25.75,40.25)	<0.001 ^b^
**Education (years)**		14 (12,16)	14.43 (12,16)	0.771 ^b^
**DD (years)**		-	4.48 (1,7)	-
**EDSS scores**		-	1.29 (0,1.5)	-
**SDMT scores**		50.42 ± 15.21	48.17 (40,57.25)	0.954 ^b^
**MMSE scores**		29.14 (28,30)	27.83 (27,29)	0.001 ^b^
**MOCA scores**		27.06 (25,29)	25.38 (24,28)	0.027 ^b^
**Volume (cm^3^)**	**CN**	6.723 ± 0.725	6.012 ± 0.991	<0.001 ^c^
	**PT**	10.235 ± 0.961	8.820 ± 1.510	<0.001 ^c^
	**GP**	3.958 ± 0.328	3.493 ± 0.457	<0.001 ^c^
	**TH**	14.461 ± 1.448	12.333 ± 1.823	<0.001 ^c^
	**TIV**	1425.024 ± 125.437	1395.152 ± 111.560	0.108 ^d^
**QSV (ppm)**	**CN**	0.035 (0.035,0.038)	0.037 (0.035,0.038)	<0.001 ^e^
	**PT**	0.057 (0.053,0.064)	0.071 (0.068,0.078)	<0.001 ^e^
	**GP**	0.020 (0.019,0.021)	0.025 (0.023,0.026)	<0.001 ^e^
	**TH**	0.094 (0.078,0.087)	0.706 (0.068,0.076)	<0.001 ^e^
**FA (mm^2^/s)**	**CN**	0.250 (0.228,0.271)	0.209 ± 0.048	<0.001 ^e^
	**PT**	0.253 ± 0.01	0.250 ± 0.018	0.214 ^e^
	**GP**	0.306 ± 0.046	0.289 ± 0.040	0.940 ^e^
	**TH**	0.311 ± 0.024	0.285 ± 0.032	0.072 ^e^
**MD (10^−3^ mm^2^/s)**	**CN**	0.616 (0.578,0.598)	0.666 (0.602,0.713)	0.007 ^e^
	**PT**	0.568 (0.553,0.581)	0.566 ± 0.002	0.020 ^e^
	**GP**	0.471 ± 0.006	0.465 (0.436,0.498)	0.075 ^e^
	**TH**	0.554 ± 0.002	0.570 (0.546,0.580)	0.189 ^e^

^a^ *p* obtained using Chi-square test. ^b^
*p* obtained using Mann-Whitney U test. ^c^
*p* obtained using univariate analysis of variance with age, sex, education duration and intracranial total volume as covariates. ^d^
*p* obtained using univariate analysis of variance with age, sex education duration as covariates. ^e^
*p* obtained using Multivariate analysis of variance with age, sex, education duration and the volumes of corresponding nucleus as covariates. normally distributed metric data is represented as $\bar{x}$ ± s, while non-normally distributed metric data is represented as M (Q1, Q3). DD, disease duration; EDSS, expanded disability status scale; MMSE, mini-mental state examination; MoCA, Montreal cognitive assessment; SDMT, symbol digit modalities test; QSV, quantitative susceptibility value; FA, fractional anisotropy; MD, mean diffusivity; CN, caudate nucleus; PT, putamen; GP, globus pallidus; TH, thalamus; TIV, total intracranial volume.

**Table 2 jpm-13-01488-t002:** Demographic and MRI quantitative metrics comparisons between MS patients and HC in study 2.

		HC (*n* = 53)	MS (*n* = 53)	*p*
Sex (male/female)		15/38	18/35	0.529 ^a^
Age (years)		36.53 ± 9.852	35.81 ± 10.077	0.712 ^b^
Education (years)		14.17 (12,16)	14.52 (15,16)	0.870 ^c^
DD (years)		-	4.8173 (1.00,8.00)	-
Volume (cm^3^)	CN	6.739 ± 0.758	5.939 ± 1.050	<0.001 ^d^
	PT	10.267 (9.460, 10.906)	8.89 (7.700, 10.181)	<0.001 ^d^
	GP	3.919 (3.751, 3.968)	3.516 (3.083, 3.867)	<0.001 ^d^
	TH	14.404 ± 1.351	12.233 ± 1.983	<0.001 ^d^
	TIV	1411.808 (1310.134, 1480.254)	1387.844 ± 17.732	0.048 ^e^
QSV (ppm)	CN	(0.0347,0.0377)	(0.0357,0.0386)	0.016 ^f^
	PT	0.058 ± 0.007	0.0718 ± 0.008	<0.001 ^f^
	GP	0.020 ± 0.002	0.0239 ± 0.002	<0.001 ^f^
	TH	(0.0766,0.0867)	(0.0678,0.0780)	0.008 ^f^
FA (mm^2^/s)	CN	(0.222,0.269)	0.202 ± 0.006	0.015 ^f^
	PT	0.250 ± 0.017	0.254 ± 0.018	0.335 ^f^
	GP	0.301 ± 0.0414	0.296 ± 0.036	0.552 ^f^
	TH	0.313 ± 0.0229	0.282 ± 0.032	0.229 ^f^
MD (10^−3^ mm^2^/s)	CN	(0.586,0.651)	(0.608,0.577)	0.031 ^f^
	PT	(0.559,0.582)	(0.542,0.577)	0.021 ^f^
	GP	(0.439,0.498)	(0.415,0.487)	0.268 ^f^
	TH	(0.548,0.571)	(0.545,0.601)	0.075 ^f^

^a^ *p* obtained using Chi-square test. ^b^
*p* obtained using independent samples *t* test. ^c^
*p* obtained using Mann–Whitney U test. ^d^
*p* obtained using univariate analysis of variance with age, sex, education duration and intracranial total volume as covariates. ^e^
*p* obtained using univariate analysis of variance with age, sex education duration as covariates. ^f^
*p* obtained using Multivariate analysis of variance with age, sex, education duration and the volumes of corresponding nucleus as covariates. normally distributed metric data is represented as $\bar{x}$ ± s, while non-normally distributed metric data is represented as M (Q1, Q3). DD, disease duration; EDSS, expanded disability status scale; MMSE, mini-mental state examination; MoCA, Montreal cognitive assessment; SDMT, symbol digit modalities test; QSV, quantitative susceptibility value; FA, fractional anisotropy; MD, mean diffusivity; CN, caudate nucleus; PT, putamen; GP, globus pallidus; TH, thalamus; TIV, total intracranial volume.

**Table 3 jpm-13-01488-t003:** Weights of brain regions contributing to classification between MS groups and HC.

	Region	Weights (%)	Cluster Size (vox)
QSV	CN	30.53	1666
	PT	25.54	2061
	GP	25.33	573
	TH	18.6	2083
FA	CN	31.38	1666
	PT	25.34	573
	GP	24.58	2061
	TH	18.7	2083
MD	CN	32.21	1666
	PT	25.89	573
	GP	21.11	2083
	TH	20.79	2061

QSV, quantitative susceptibility value; FA, fractional anisotropy; MD, mean diffusivity; CN, caudate nucleus; PT, putamen; GP, globus pallidus; TH, thalamus.

## Data Availability

Data available on request due to privacy. The data presented in this study are available on request from the corresponding author.

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
