# Peer review of "Multimodal Investigation of Deep Gray Matter Nucleus in Patients with Multiple Sclerosis and Their Clinical Correlations: A Multivariate Pattern Analysis Study"

_jpm, 2023, doi:10.3390/jpm13101488_

Round 1
Reviewer 1 Report
This report detailed a multimodal, forward-thinking machine learning approach to investigate microstructural alterations in deep grey matter of patients with multiple sclerosis (MS). The article ates that ‘one hundred and fifteen MS patients 13 And seventy-one healthy controls (HC) underwent brain MRI.
Multivariate pattern analysis, based on a machine-learning algorithm, was applied to explore affected tissue regions of interest. A focus was applied to correlation between MRI quantitative metrics and clinical neurological scores.
The limitation of this study is clearly declared by the authors being sample size and future work will be an interesting read especially as mentioned; correlation between lesion burden and clinical neurological scales.
Reviewer 2 Report
The paper entitled "Multimodal Investigation of Deep Gray Matter Nucleus in Patients with Multiple Sclerosis and Their Clinical Correlations: A Multivariate Pattern Analysis Study" brings valuable information about the MRI characteristics of multiple sclerosis patients.
Observations
Introduction
Please write in details the mechanisms of iron deposition in brain in multiple sclerosis patients, consecutive to neurodegeneration and to its consequences.
Since you found the atrophy of DGM, please write more about the mechanism of DGM atrophy.
Material and methods
Please write a table with comparative demographics characteristic of control group and patients group, to be more visible. What was the average of control group age? You need to show if there where statistical differences between these groups's age.
Please also indicate the multiple sclerosis type in the study group. How many patients had CIS, RRMS, PPMS and SMPS? These forms are important for clinical evolution you assessed, for demyelination process, for brain lesions progression/evolution and neurodegenerative mechanism.
Conclusions
Please write a separate chapter about conclusions of your study.
